# Impact of Acute Aerobic Exercise on Genome-Wide DNA-Methylation in Natural Killer Cells—A Pilot Study

**DOI:** 10.3390/genes10050380

**Published:** 2019-05-19

**Authors:** Alexander Schenk, Christine Koliamitra, Claus Jürgen Bauer, Robert Schier, Michal R. Schweiger, Wilhelm Bloch, Philipp Zimmer

**Affiliations:** 1Department of Molecular and Cellular Sport Medicine, Institute for Cardiology and Sports Medicine, German Sport University Cologne, Am Sportpark Müngersdorf 6, 50933 Cologne, Germany; a.schenk@dshs-koeln.de (A.S.); c.koliamitra@dshs-koeln.de (C.K.); w.bloch@dshs-koeln.de (W.B.); 2Department of Anesthesiology and Intensive Care Medicine, University Hospital of Cologne, Kerpener Straße 62, 50937 Cologne, Germany; cjbatwork@googlemail.com (C.J.B.); robert.schier@uk-koeln.de (R.S.); 3Department of Epigenetics and Tumor Genetics, Medical Faculty, University of Cologne, Kerperner Straße 62, 50937 Cologne, Germany; mschweig@uni-koeln.de; 4Institute of Human Genetics, University Hospital of Cologne, Kerpener Straße 62, 50937 Cologne, Germany; 5Center for Molecular Medicine Cologne, Kerperner Straße 62, 50937 Cologne, Germany

**Keywords:** natural killer cell, NK, epigenetics, DNA-methylation, exercise, sport

## Abstract

Natural Killer (NK-) cells reveal a keen reaction to acute bouts of exercise, including changes of epigenetic modifications. So far, exercise-induced alterations in NK-cell DNA-methylation were shown for single genes only. Studies analyzing genome-wide DNA-methylation have used conglomerates like peripheral blood mononuclear cells (PBMCs) rather than specific subsets of immune cells. Therefore, the aim of this pilot-study was to generate first insights into the influence of a single bout of exercise on genome-wide DNA-methylation in isolated NK-cells to open the field for such analyses. Five healthy women performed an incremental step test and blood samples were taken before and after exercise. DNA was isolated from magnet bead sorted NK-cells and further analyzed for global DNA-methylation using the Infinium MethylationEPIC BeadChip. DNA-methylation was changed at 33 targets after acute exercise. These targets were annotated to 25 genes. Of the targets, 19 showed decreased and 14 increased methylation. The 25 genes with altered DNA-methylation have different roles in cell regulation and differ in their molecular functions. These data give new insights in the exercise induced regulation of NK-cells. By using isolated NK-cells, exercise induced differences in DNA-methylation could be shown. Whether or not these changes lead to functional adaptions needs to be elucidated.

## 1. Introduction

Natural killer (NK-) cells are a distinct cell population of the innate immune system that build the first line of defense against virus-infected and neoplastic cells [1]. NK-cells are characterized by their rapid activation through germ line encoded receptors [2] and their ability to discriminate between healthy and diseased cells. NK-cells show a keen reaction to acute bouts of exercise, including changes in circulating NK-cell numbers and NK-cell function [3]. Circulating NK-cell numbers increase during exercise and decrease immediately after cessation of exercise below pre-exercise levels for up to 48 h [4]. Furthermore, the NK-cell pool can be divided into the more regulatory CD56^bright^ subset and the pronounced cytotoxic CD56^dim^ subset. As shown by Bigley et al. [5], exercise predominantly affects the redistribution of CD56^dim^ NK-cells, leading to changes in the distribution of NK-cell subsets. Data for NK-cell function is inconsistent but a tendency regarding higher NK-cell function after intensive exercise is reported [3].

Research of the past decade suggests that several short- and long-term adaptions of various tissues in response to exercise are based on epigenetic modifications [6,7,8,9], such as DNA-methylation, histone modification and microRNA expression [10]. In view of immune cells, results from Nakajima et al. [7] and Denham et al. [11] suggested gene-specific changes of DNA-methylation in peripheral blood mononuclear cells (PBMCs) and leukocytes after interval training interventions. Regarding acute exercise, Robson-Ansley et al. [12] found no effects on global DNA-methylation in PBMCs. However, analyzing PBMCs has some major limitations. First, PBMCs represent an agglomerate of several immune cell subsets which have distinct epigenetic patterns. Second, acute exercise and training interventions alter the number and proportions of immune cell subsets. Thereby, detected changes of DNA-methylation may be driven by changes in PBMC composition rather than by changes on the epigenome. Hence, analyzing specific immune cell subsets would be more accurate. Another attempt to overcome these problems was conducted by Horsburgh et al. [13]. PBMCs were cultivated in vitro with plasma collected before and after an intensive exercise bout. They found the nuclear expression of DNA methyltransferase (DNMT) 3B to be decreased after exercise. DNMT3B is one of the DNMTs, accounting for de novo DNA-methylation and therefore changes in DNA-methylation were suggested but unfortunately not tested [13].

Using isolated NK-cells, we [14] have shown that a long-lasting acute bout of endurance exercise (half marathon) did not alter global DNA-methylation. However, this was only tested by global immunocytochemistry. Nevertheless, antibody detected global histone acetylation at the fifth lysine-residue of histone 4 (H4K5) increased after the run and was further accompanied by an elevated gene expression of the activating NK-cell receptor NKG2D. Interestingly, gene expression of NKG2D has previously been described to be regulated by epigenetic modifications [15]. Furthermore, we have recently shown that an acute bout of exercise decreased the promoter DNA-methylation of the activating killer immunoglobulin-like receptor (KIR) KIR2DS4 and increased the corresponding gene expression [16]. In contrast, neither promoter DNA-methylation nor gene expression of the inhibitory *KIR3DL1* gene was affected. In this trial, DNA-methylation was assessed using the high-resolution method targeted deep amplicon bisulfite sequencing (TDBS).

Since immunohistochemically antibody-based assessment of DNA-methylation is relatively vague, and TDBS does not allow the analysis of a wide range of genes, this pilot-study aims to generate first insights into the influence of a single bout of exercise on gene specific DNA-methylation using a microarray approach in isolated NK-cells and open the field for such analyses.

## 2. Experimental Section

This study was performed in accordance with the declaration of Helsinki and was approved by the ethics committee of the University Hospital of Cologne (IRB#13-274). Five healthy women (age 61.4 ± 8.0 years) were recruited by regional newspaper announcement between December 2014 and September 2015 and provided written informed consent. The term “healthy” was defined as no acute or chronic diseases. Any kind of drug intake was an exclusion criterion. For analyzing the effects of a single bout of exercise on the methylome of NK-cells, venous blood samples were collected before (t0) and 1 min after (t1) an incremental step test on a bicycle ergometer.

### 2.1. Incremental Step Test

The incremental step test was performed on a bicycle ergometer (Ergoline GmbH, Bitz, Germany) with a spirometry analysis (Cortex Biophysik GmbH, Leipzig, Germany). The incremental step test began with a 1 min rest measurement, followed by a 3 min warm-up at 50 watts power output and an increase of 25 watts every 2 min until exhaustion (Respiratory quotient >1). Heart rate (Promedia Medizintechnik, Siegen, Germany) and self-perceived exhaustion (Borg scale) were assessed in each intensity step.

### 2.2. Blood Sampling and NK-Cell Isolation

Blood samples were collected before (t0) and 1 min after (t1) the incremental step test. Blood samples were used for isolation of peripheral blood mononuclear cells (PBMCs) with a lymphocyte separation medium (promo Cell, Heidelberg, Germany) based density gradient centrifugation. Isolated PBMCs were used for a magnetic bead associated negative separation of NK-cells (EasySep™ Human NK Cell Enrichment Kit; STEMCELL Technologies Germany GmbH, Cologne, Germany) according to the manufacturer’s protocol.

### 2.3. DNA Isolation

DNA was isolated using a column-based isolation (RNA/DNA/Protein Purification Plus Kit; Norgen Biotek Corp., Thorold, ON, Canada) according to the manufacturer’s protocol.

### 2.4. Genome-Wide DNA-Methylation

Genome-wide DNA-methylation was assessed with the Infinium^®^ MethylationEPIC BeadChip (Illumina, San Diego, CA, USA) according to the manufacturer´s protocol. Methylation data was calculated with Genome Studio 2011.1 containing the methylation module (Illumina, San Diego, CA, USA). Methylation was displayed in β-values. The data discussed have been deposited in NCBI’s Gene Expression Omnibus and are accessible through GEO Series accession number GSE129376 (https://www.ncbi.nlm.nih.gov/geo/query/acc.cgi?acc=GSE129376).

### 2.5. Gene Ontology

Functional annotation of genes showing changes in DNA-methylation was performed using the DAVID Bioinformatics Resources [17].

### 2.6. Flow Cytometry

To control for changes of the NK-cell subsets distribution, isolated PBMCs were used for flow cytometry analysis. PBMCs were stained with anti-CD3 APC-C7 and anti-CD56 PE (BD Bioscience, Heidelberg, Germany). Flow cytometry was performed on a BD FACS Array (BD Bioscience, Heidelberg, Germany) and NK-cells were gated as CD3− and CD56+. Furthermore, CD56^bright^ and CD56^dim^ NK-cells were gated and presented as percentage of the NK-cell pool.

### 2.7. Statistics

Genome Studio was used to perform a differential methylation analysis, to detect differences between before (t0) and after (t1) acute exercise. The illumina custom model, including consideration of the false discovery rate (FDR) by *p*-value correction according to Benjamini-Hochberg [18], was used to calculate the differential methylation. FDR corrected DiffScores were computed, with a DiffScore ≥ |13| ≈ *p* ≤ 0.05 as described by Tremblay et al. [19]. In brief, the detection *p*-value indicated whether the sample signal was distinguishable from the negative control. Targets indicating a detection *p* > 0.05 in more than 10% of the samples were excluded from analyses. Flow cytometry data was compared using a Wilcoxon signed-rank test. The level of significance was set at *p* ≤ 0.05.

## 3. Results

### 3.1. Participants Characteristics

To analyze the influence of a single bout of acute exercise, five healthy women were recruited with an age of 61.4 ± 8.0 years. The incremental step test revealed a peak oxygen uptake (VO_2peak_) of 32.4 ± 3.8 mL/min/kg with a maximal power output of 165.0 ± 13.7 Watt. Detailed characteristics of the participants are shown in Table 1.

### 3.2. Genome-Wide DNA-Methylation

Analysis with the Infinium MethylationEPIC BeadChip revealed 864,614 targets of a total of 865,918 (99.8%) that could be detected with a detection *p* ≤ 0.05. Comparing DNA-methylation before (t0) and after (t1) acute exercise revealed 33 targets being changed with a DiffScore ≥ |13| (0.004%). Within the 33 targets affected by acute exercise, 25 were annotated to a gene. As shown in Table 2, in 19 of the 33 targets (57.6%) DNA-methylation was decreased after exercise, whereas 14 targets (42.4%) show increased DNA-methylation. A total of 10 targets (30.3%) were found in a gene body and 8 targets (24.2%) had no gene annotation. Table 2 presents a complete list of targets with a changed methylation, including gene information and β-values.

### 3.3. Gene Ontology

DAVID Bioinformatics Resources [17] was used for functional annotation of the 25 genes (Table 3). It is noteworthy that *FASLG* is the gene with the most functional annotation. It belongs to the plasma membrane and is involved in regulation of transcription.

### 3.4. Distribution of NK-Cell Subsets CD56^bright^ and CD56^dim^

Flow cytometry was used to control blood samples for changes in NK-cell subset distribution. The percent of CD56^bright^ NK-cells within the NK-cell pool, decreased from 14.91% ± 4.64% to 10.38% ± 3.20% comparing before and after acute exercise. Accordingly, the percent of CD56^dim^ NK-cell increased from 84.63% ± 4.78% to 89.12% ± 3.57%. Nevertheless, both changes did not reach statistical significance (*p* = 0.080, *p* = 0.080).

## 4. Discussion

### 4.1. Genome-Wide DNA-Methylation

Investigating the effect of exercise interventions on DNA-methylation in specific immune cells is a novel research field and literature in this context is limited to studies analyzing PBMCs [7,11,12,13] or whole blood samples [20]. To our knowledge, this is the first study investigating the influence of an acute bout of endurance exercise on genome-wide DNA-methylation on a single cytosine guanine dinucleotide (CpG) level in isolated NK-cells. The results reveal 33 targets being differentially methylated after exercise, with 25 genes corresponding to these targets. Overall, 14 targets show increased DNA-methylation, whereas 19 targets show a decreased methylation.

Our results reveal a fast adaption of NK-cells to an acute bout of endurance exercise by changes of DNA-methylation at 33 targets. So far, DNA-methylation was thought to be a more stable epigenetic modification when compared to histone modifications or miRNAs. Nevertheless, our results indicate a more dynamic function of DNA-methylation in response to external stimuli, e.g., exercise, and this hypothesis is supported by recent research. It was shown that a single bout of exercise changes promoter DNA-methylation in muscle [21] and adipose tissue [22]. In view of blood cells, Robson-Ansley et al. [12] showed keener changes in DNA-methylation immediately after a single bout of exercise compared to a 24 h post-exercise assessment. The authors described a correlation of changes in DNA-methylation of the influenced genes with interleukin 6 (IL-6) levels, indicating changes of cytokine levels as a mediator of changes in DNA-methylation. Horsburgh et al. [13] found a decrease of nuclear concentrations of DNA methyltransferases (DNMTs) of PBMCs after in vitro cultivation with exercise preconditioned plasma. Furthermore, in a recent study we showed that acute exercise resulted in promoter demethylation of the activating NK-cell receptor KIR2DS4 [16]. In the same study also the inhibiting KIR3DL1 was investigated, showing no changes after acute exercise. Interestingly, the presented microarray attempt did not show changes at the *KIR2DS4*-gene. Comparing the *KIR2DS4* and *KIR3DL1* genes in both attempts, we found 6 CpGs (4 in the *KIR2DS4* gene and 2 in the *KIR3DL1* gene), which were analyzed in both attempts (TDBS and EPIC). All 6 commonly analyzed CpGs remained unaltered when comparing both methods and the ones that were changed in the previous study [16] were unfortunately not analyzed within the microarray. It is emphasized that a combination of methods is needed to get detailed information about changes in DNA-methylation.

Analysis of functional annotation reveals *FASLG* with the most annotations. *FASLG* codes for a ligand of the FAS-receptor and induces apoptosis upon binding, one mechanism of NK-cell cytotoxicity. Our results reveal a slightly reduced DNA-methylation of *FASLG* in one CpG within the first exon after acute exercise. As shown by Mooren et al. [23], gene expression of FASLG in leukocytes is not changed immediately after exercise, but it is increased 3 h later. Whether or not there is an association between changes in DNA-methylation and gene expression remains unclear and should be investigated in following studies.

### 4.2. Distribution of NK-Cell Subsets

The NK-cell subsets CD56^bright^ and CD56^dim^ reveal different functional properties [24] and may also differ in their epigenetic imprinting. It is well described that the NK-cell numbers within the blood stream rapidly increase during exercise and decrease below pre-exercise levels (reviewed by [24]). Moreover, exercise redeploys preferentially CD56^dim^ NK-cells [5], possibly leading to a change in subset proportions and therefore changes in the epigenetic pattern of the entire NK-cell pool. In contrast, we found no statistically significant changes in the NK-cell populations before and directly after exercise. This could be due to the short duration of the exercise bout. While the exercise bout in our study lasted about 15 min, Campbell and Turner [25] describe exercise bouts of 45–60 min duration being characterized by changes in blood lymphocyte numbers. Therefore, it is hypothesized that the short duration of this intensive exercise bout explains the absence of explicit changes in NK-cell subset distribution. Fortunately, this data also supports the finding that changes in DNA-methylation are not driven by changes in NK-cell subsets. Even if the short intensive bout of exercise was not sufficient to induce changes in NK-cell subset distribution, it was sufficient to induce changes in DNA-methylation. It is suggested that a longer bout would induce more pronounced changes, so future studies should include analysis of a single cell population and use longer bouts of exercise.

### 4.3. Limitations

The results of the presented study should be considered within the context of its limitations. Chip based microarrays for DNA-methylation analyses are a state-of-the-art method and allow the quantification of DNA-methylation at a vast number of single CpGs in parallel. Especially, the Illumina MethylationEPIC BeadChip is described as a robust and reliable platform [26]. Nevertheless, an a priori determined selection of CpGs may also discriminate and possible changes remain undetected. The use of whole genome bisulfate sequencing may provide detailed information of DNA-methylation based gene regulation by measuring all CpGs and all genes. A major limitation is the small sample size of five participants. Nevertheless, it is the first study investigating genome-wide DNA-methylation with a microarray design in isolated NK-cells in the context of physical exercise. In contrast to previous studies investigating cell conglomerates like PBMCs, this attempt produced more precise results for the regulation of a specific cell type. Since NK-cells are further divided into subgroups, analyses of sorted NK-cell subpopulations (CD56^dim^ and CD56^bright^) provided even more detailed information of differential regulation of the subgroups. However, the amount of DNA isolated from NK-cell subpopulations would be a bottle neck for those analyses. Moreover, the timing of blood sampling after exercise affected the composition of circulating leukocytes and was standardized precisely. By using venipuncture, each blood sampling could be subject to a minor delay and therefore may have influenced the results. Since a change in NK-cell subset proportions was absent, exercise bouts of longer durations should be used to increase the influence of exercise on the NK-cell pool. Furthermore, more measurement time points could give information about the kinetics of exercise induced changes of DNA-methylation and should be considered in following studies. Gene expression of affected genes is not measured. Consecutive studies are warranted, analyzing whether changes in DNA-methylation correspond to changes in gene expression of the affected genes.

## 5. Conclusions

In conclusion, this is the first study showing effects of a single bout of endurance exercise on genome-wide DNA-methylation in isolated NK-cells. Acute exercise affects the functional genome of NK-cells by changes in DNA-methylation. In order to investigate such changes, it is proposed to use specific cell types rather than conglomerates like PBMCs. More research is needed to combine changes in the functional genome of NK-cells with changes in NK-cell function.

## Figures and Tables

**Table 1 genes-10-00380-t001:** Participant characteristics.

Parameter	Mean	Standard Deviation
Age [years]	61.4	8.0
Weight [kg]	60.2	4.0
Height [cm]	166.2	4.1
BMI [kg/m^2^]	20.0	1.5
Waist circumference [cm]	84.2	5.5
Maximal power output [Watt]	165.0	13.7
VO_2peak_ [mL/min/kg]	32.4	3.8

**Table 2 genes-10-00380-t002:** Detailed presentation of targets affected by acute exercise. Genomic location of targets is given by the chromosome and location, as well as its orientation to the gene (refgene group). If a gene is annotated to the target cytosine guanine dinucleotide (CpG), the refgene name is presented. The difference in DNA-methylation for each target is shown as Δβ-value and the corresponding DiffScore given. The DiffScore is a transformation of the *p*-value and indicates differences between both time points. A DiffScore of ≥ |13| represents a *p* ≤ 0.05. Presented is the false discovery rate corrected DiffScore.

Target ID	Chromosome	Location	Refgene Group	Refgene Name	Δβ-Value	DiffScore
cg03347334	18	55829444	Body	*NEDD4L*	−0.16	−60.15
cg05476733	11	128477400			−0.13	−53.12
cg23944405	11	30602030	1stExon	*MPPED2*	−0.13	−42.75
cg18139862	3	48344301	TSS1500	*NME6*	−0.14	−41.93
cg21899777	22	46771084	Body	*CELSR1*	−0.10	−37.09
cg19360943	12	6762431	Body	*ING4*	−0.11	−36.96
cg22942704	1	20813574	TSS1500	*CAMK2N1*	−0.15	−35.73
cg13565400	11	73882059	1stExon	*C2CD3; PPME1*	−0.10	−23.19
cg02295170	6	130718139	5′UTR	*TMEM200A*	−0.09	−22.73
cg24226193	7	28191663	Body	*JAZF1*	−0.11	−22.50
cg05119374	6	32399399			−0.09	−20.57
cg27114965	3	57614340	Body	*DENND6A*	−0.10	−19.06
cg00268500	2	64067540	TSS1500	*UGP2*	−0.09	−17.98
cg20481642	2	166060635	TSS200	*SCN3A*	−0.08	−16.45
cg15729230	1	172628514	1stExon	*FASLG*	−0.10	−15.97
cg03997458	10	125207543			−0.08	−15.61
cg01379853	19	6239836	Body	*MLLT1*	−0.10	−13.46
cg21895314	21	44593708			−0.10	−13.28
cg20339715	8	27757965	Body	*SCARA5*	−0.11	−13.20
cg00835758	14	35550189	TSS200	*LOC101927178; FAM177A1*	0.13	15.61
cg21646955	6	35108921	Body	*TCP11*	0.08	16.18
cg25540806	4	90815778	TSS1500	*MMRN1*	0.11	16.61
cg22758714	4	190942739			0.12	17.75
cg07675898	11	41681482			0.08	19.76
cg06716138	8	124857543			0.12	20.77
cg11066566	3	17783373	TSS1500	*TBC1D5*	0.10	22.87
cg14678442	17	54672540	1stExon	*NOG*	0.10	23.97
cg17395184	15	42750462	TSS1500	*ZFP106*	0.13	29.99
cg02771649	1	31474920	Body	*PUM1*	0.13	31.01
cg20381404	5	34008215	5′UTR	*AMACR*	0.09	37.09
cg03681640	2	6647183			0.18	51.10
cg02270786	1	45474858	Body	*HECTD3*	0.11	61.64
cg06095510	17	56764569	5′UTR	*TEX14*	0.17	314.97

**Table 3 genes-10-00380-t003:** Functional Annotation. The functional annotation was performed using the DAVID software. Category gives the database used by the DAVID software. Term gives the functional annotation.

Category	Term	Genes
UP_SEQ_FEATURE	Mutagenesis site	*FASLG, NEDD4L, PUM1, SCN3A, UGP2*
UP_KEYWORDS	Ubl conjugation	*FASLG, NEDD4L, SCN3A, ZNF106*
GOTERM_CC_DIRECT	GO:0005886~plasma membrane	*CELSR1, FASLG, NEDD4L, SCN3A*
UP_KEYWORDS	Metal-binding	*ING4, JAZF1, MPPED2, NME6, UGP2, ZNF106*
UP_KEYWORDS	Zinc-finger	*ING4, JAZF1, ZNF106*
UP_KEYWORDS	Zinc	*ING4, JAZF1, ZNF106*
UP_KEYWORDS	Nucleus	*FASLG, ING4, JAZF1, MLLT1, ZNF106*
UP_SEQ_FEATURE	Glycosylation site:N-linked (GlcNAc…)	*CELSR1, FASLG, MMRN1, NOG, SCARA5, SCN3A, TMEM200A*
UP_KEYWORDS	Glycoprotein	*CELSR1, FASLG, MMRN1, NOG, SCARA5, SCN3A, TMEM200A*
UP_SEQ_FEATURE	Disulfide bond	*CELSR1, MMRN1, NOG, FASLG, SCARA5*
UP_KEYWORDS	Cell membrane	*CAMK2N1, CELSR1, FASLG, SCARA5, SCN3A*
GOTERM_CC_DIRECT	GO:0005887~integral component of plasma membrane	*CELSR1, FASLG, SCARA5*
UP_KEYWORDS	Disulfide bond	*CELSR1, FASLG, MMRN1, NOG, SCARA5*
GOTERM_CC_DIRECT	GO:0005576~extracellular region	*FASLG, MMRN1, NOG*
UP_SEQ_FEATURE	Topological domain:Extracellular	*CELSR1, FASLG, SCARA5, TMEM200A*
UP_KEYWORDS	Secreted	*FASLG, MMRN1, NOG*
UP_SEQ_FEATURE	Transmembrane region	*CELSR1, FASLG, SCARA5, SCN3A, TCP11, TMEM200A*
UP_SEQ_FEATURE	Signal peptide	*AMACR, CELSR1, MMRN1, NOG*
UP_SEQ_FEATURE	Topological domain:Cytoplasmic	*CELSR1, FASLG, SCARA5, TMEM200A*
UP_KEYWORDS	Transmembrane helix	*CELSR1, FASLG, SCARA5, SCN3A, TCP11, TMEM200A*
UP_KEYWORDS	Membrane	*CAMK2N1, CELSR1, FASLG, SCARA5, SCN3A, TBC1D5, TCP11, TMEM200A*
UP_KEYWORDS	Transmembrane	*CELSR1, FASLG, SCARA5, SCN3A, TCP11, TMEM200A*
GOTERM_CC_DIRECT	GO:0005886~plasma membrane	*CELSR1, FASLG, NEDD4L, SCN3A*
GOTERM_CC_DIRECT	GO:0016021~integral component of membrane	*CELSR1, FASLG, SCN3A, TCP11, TMEM200A*
UP_KEYWORDS	Signal	*CELSR1, MMRN1, NOG*
GOTERM_CC_DIRECT	GO:0005634~nucleus	*FASLG, HECTD3, ING4, JAZF1, MLLT1, NEDD4L, UGP2*
UP_KEYWORDS	Transcription regulation	*FASLG, JAZF1, MLLT1*
UP_KEYWORDS	Transcription	*FASLG, JAZF1, MLLT1*
UP_KEYWORDS	Nucleus	*FASLG, ING4, JAZF1, MLLT1, ZNF106*

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
