# Peer review of "Impact of Acute Aerobic Exercise on Genome-Wide DNA-Methylation in Natural Killer Cells—A Pilot Study"

_genes, 2019, doi:10.3390/genes10050380_

Reviewer 1 Report

The study assesses the impact of acute aerobic exercise on global DNA methylation in NK cells in peripheral blood.

We believe that given the very low number of subjects (5 individuals) and the very high number of targets (846 614 targets),

- the detailed statistical procedure by which FDR has been taken into consideration, should be described

- the analytical performance of the global DNA analysis method, in terms of repeatability should be assessed and reported. For example, when we run the same sample five times, and we consider the first samples as a "baseline", how many targets  appear to be significantly different from the baseline in samples 2,3,4, and 5?

- Technical duplicates may be needed

- Confirmation with an independent cohort of samples is needed (even a small one, but an independent one)

- Assessment of the linearity of the potential DNA methylation-related response to the length of the acute exercise would bring a convincing argument for the specificity of such a response.

In the statistics section, it is mentioned that p-value indicated whether sample signal was distinguishable from the background. Could the authors specify what background? Do authors refer to the baseline measurements?

Author Response

Reviewer 1

Dear Reviewer,

We would like to thank you for your constructive criticism. We agree with your comments and tried to address them appropriately. We hope we could improve the manuscript with the comments of both reviewers. Nevertheless, this concept paper presents a pilot study which should open the field of exercise immunology to such kind of analyses in isolated cell populations. Therefore, this pilot study should be seen as a basis for following RCTs.

Here you find the point-by-point comments:

The study assesses the impact of acute aerobic exercise on global DNA methylation in NK cells in peripheral blood.

We believe that given the very low number of subjects (5 individuals) and the very high number of targets (846 614 targets),

- the detailed statistical procedure by which FDR has been taken into consideration, should be described

The calculation of FDR is a p-value correction according to the method of Benjamini and Hochberg. We specified this in the manuscript.

- the analytical performance of the global DNA analysis method, in terms of repeatability should be assessed and reported. For example, when we run the same sample five times, and we consider the first samples as a "baseline", how many targets  appear to be significantly different from the baseline in samples 2,3,4, and 5?

- Technical duplicates may be needed

For both points: Unfortunately, we do not have the material, to perform the analyses that often with the same samples. But we included literature describing the used assays as very robust and reliable.

- Confirmation with an independent cohort of samples is needed (even a small one, but an independent one)

We understand the question for a passive control group for confirmation of the results. Nevertheless, in the context of acute exercise studies the need of a passive control group is discussed controversial. It is argued that the short duration of the acute exercise bouts would not induce changes in a passive control group, so the baseline values could be used as a “control” for the changes. Because this is a pilot study and preliminary results, we decided not to use a passive control group. In following studies with a higher number of cases, a passive control group should be included.

- Assessment of the linearity of the potential DNA methylation-related response to the length of the acute exercise would bring a convincing argument for the specificity of such a response.

In this pilot study the overall effect of single bout of exercise was investigated. To investigate the linearity of DNA methylation responses is another research question that we did not want to address in this study. In a first step we wanted to investigate if there are changes, before thinking about the kinetics of changes. Therefore, we agree that this research question should be addressed in following studies and state this in the discussion.

In the statistics section, it is mentioned that p-value indicated whether sample signal was distinguishable from the background. Could the authors specify what background? Do authors refer to the baseline measurements?

We specified this information. The detection p-value compares the sample to a negative control. Insignificant samples are not distinguishable from a negative control and are removed from the analyses. This is performed at the end of the analyses, even after consideration of FDR.

Reviewer 2 Report

The article was an interesting read and I was not aware that NK-cell numbers increase post exercise (i.e. > 45 min).

In the current study exercise of 15 min was used and there was no difference in NK-cell numbers. However, there was a difference genome methylation (i.e. on 25 genes). One of the obvious questions is whether an analysis of five participants is enough to draw these conclusions and I would think it is not. Can the authors perform a power analysis showing that the sample number was sufficient to support their findings ?

I’m also wondering how genome methylation of NK-cells changes when the immune system is stimulated and how this compares to their findings ? How many genes get differentially methylated in both cases ?

Another obvious question is of course whether or not the identified differences in methylation have any functional consequences ?

Author Response

Reviewer 2

Dear Reviewer,

We would like to thank you for your constructive criticism. We agree with your comments and tried to address them appropriately. We hope we could improve the manuscript with the comments of both reviewers. Nevertheless, this concept paper presents a pilot study which should open the field of exercise immunology to such kind of analyses in isolated cell populations. Therefore, this pilot study should be seen as a basis for following RCTs.

Here you find the point-by-point comments:

The article was an interesting read and I was not aware that NK-cell numbers increase post exercise (i.e. > 45 min).

This information is stated in the new review of Campbell and Turner. We specified this information by stating more precise that it is 45 Min duration of exercise.

 In the current study exercise of 15 min was used and there was no difference in NK-cell numbers. However, there was a difference genome methylation (i.e. on 25 genes). One of the obvious questions is whether an analysis of five participants is enough to draw these conclusions and I would think it is not. Can the authors perform a power analysis showing that the sample number was sufficient to support their findings ?

We agree with the reviewer´s comment, that the results are only preliminary and should be approved with following RCTs. However, such pilot data are necessary to get an idea of potential effects and open the field for further research.

I’m also wondering how genome methylation of NK-cells changes when the immune system is stimulated and how this compares to their findings ? How many genes get differentially methylated in both cases ?

 We compared our results with the results of Wiencke et al. 2016, who investigated the methylation profile of activated NK-cells. We did not found any accordance with these results. However, the authors used an artificial activation of NK-cells that might be not comparable with the influence of exercise. It could also be argued that exercise does not lead to an activation of NK-cells, rather than to a higher degree of activatability.

Another obvious question is of course whether or not the identified differences in methylation have any functional consequences ?

In this pilot study it was not the research question if the acute bout of exercise will lead to functional changes. We agree that following studies should include functional analyses to evaluate the clinical relevance of changes in DNA-methylation.

Round  2

Reviewer 1 Report

the fact the this is a pilot study that provides indications that are meant to open the field of investigations in this area should be clearly stated upfront.

Author Response

Pont 1: the fact the this is a pilot study that provides indications that are meant to open the field of investigations in this area should be clearly stated upfront.

Dear Reviewer,

thank you for this comment. We precised the goal of this publication at the end of the introduction accordung to your suggestion. Furthermore, we also added this into the abstract.